# Investigation of the Effects of a Novel NOX2 Inhibitor, GLX7013170, against Glutamate Excitotoxicity and Diabetes Insults in the Retina

**DOI:** 10.3390/ph17030393

**Published:** 2024-03-19

**Authors:** Stavroula Dionysopoulou, Per Wikstrom, Erik Walum, Spiros Georgakis, Kyriaki Thermos

**Affiliations:** 1Department of Pharmacology, School of Medicine, University of Crete, 70013 Heraklion, Crete, Greece; med2p1080136@med.uoc.gr; 2Glucox Biotech AB, S-17997 Stockholm, Sweden; per.wikstrom@glucoxbiotech.com (P.W.); walumerik@gmail.com (E.W.); 3Laboratory of Rheumatology, Autoimmunity and Inflammation, School of Medicine, University of Crete, 70013 Heraklion, Crete, Greece; georgakis.spyridon@chuv.ch

**Keywords:** NADPH oxidase, NOX2 inhibition, glutamate excitotoxicity, early-stage diabetic retinopathy, neurodegeneration, neuroinflammation, neuroprotection, excitatory amino acid transporter (EAAT1)

## Abstract

Glutamate excitotoxicity and oxidative stress represent two major pathological mechanisms implicated in retinal disorders. In Diabetic Retinopathy (DR), oxidative stress is correlated to NADPH oxidase (NOX), a major source of Reactive Oxygen Species (ROS), and glutamate metabolism impairments. This study investigated the role of NOX2 and the novel NOX2 inhibitor, GLX7013170, in two models of a) retinal AMPA excitotoxicity [AMPA+GLX7013170 (10^−4^ M, intravitreally)] and b) early-stage DR paradigm (ESDR), GLX7013170: 14-day therapeutic treatment (topically, 20 μL/eye, 10 mg/mL (300 × 10^−4^ M), once daily) post-streptozotocin (STZ)-induced DR. Immunohistochemical studies for neuronal markers, nitrotyrosine, micro/macroglia, and real-time PCR, Western blot, and glutamate colorimetric assays were conducted. Diabetes increased NOX2 expression in the retina. NOX2 inhibition limited the loss of NOS-positive amacrine cells and the overactivation of micro/macroglia in both models. In the diabetic retina, GLX7013170 had no effect on retinal ganglion cell axons, but reduced oxidative damage, increased Bcl-2, reduced glutamate levels, and partially restored excitatory amino acid transporter (EAAT1) expression. These results suggest that NOX2 in diabetes is part of the triad, oxidative stress, NOX, and glutamate excitotoxicity, key players in the induction of DR. GLX7013170 is efficacious as a neuroprotective/anti-inflammatory agent and a potential therapeutic in retinal diseases, including ESDR.

## 1. Introduction

Oxidative stress and glutamate excitotoxicity represent two major pathogenic mechanisms involved in a plethora of Central Nervous System (CNS) disorders, including various pathologies that affect the retina. Many years of research have corroborated the complexity of these mechanisms, independently, and their crucial impact on neural tissue. In addition, an interplay between them has been identified. In retinal pathologies such as Diabetic Retinopathy (DR), understanding the interconnection between different pathogenic mechanisms may contribute to a better comprehension of the basis of the disease and the development of novel therapeutics able to cope with multiple pathological components and provide an improved therapeutic impact.

The term “oxidative stress” refers to the imbalance between the production of free radicals and their removal from the endogenous antioxidant systems that could lead to severe cell damage and tissue injury [1,2]. The two major sources of Reactive Oxygen Species (ROS) are the mitochondria and the Nicotinamide Adenine Dinucleotide Phosphate (NADPH) oxidase (NOX) system [3]. In mitochondria, ROS are generated during oxidative phosphorylation by the electron transport chain. However, NADPH oxidases are multi-subunit enzymatic complexes, which catalyze the transfer of one electron from NADPH to molecular oxygen, leading to the production of superoxide (O2^•−^), which can further react and give rise to other free radicals [4]. To this day, seven isoforms of NADPH oxidases have been identified, constituting the “NOX family”: NOX1–NOX5, DUOX1, and DUOX2, with NOX2 being the first identified and described [5].

In the retina, NOX1, 2, and 4 are expressed in retinal neurons, including retinal ganglion cells (RGCs) and in micro/macroglia and pericytes [6]. The main source of ROS in the retina is the microglia, a significant number of which are derived from microglia-expressing NADPH oxidases [6]. Moreover, in humans, all known NOX isoforms, including NOX5 and DUOX1/2, are expressed in retinal endothelial cells, where NOX-derived ROS participate in the functional regulation of retinal vasculature [6,7]. Under various retinal pathological conditions, like oxygen-induced retinopathy [8] and diabetic retinopathy [9,10,11], the expression profile of NOX exhibits significant changes, leading to increased production of ROS and the subsequent development of neurotoxic phenomena in the retina. Thus, molecules and strategies aimed to inhibit the activity of NOX isoforms are of particular scientific interest, since there is an essential need for new potential therapeutics for retinal pathologies.

The excitotoxic cascade begins as a result of an ischemia-induced increase in glutamate levels, the subsequent persistent activation mostly of calcium ion (Ca^2+^)permeable N-methyl-D-aspartate (NMDA) glutamate, but also of Ca^2+^ permeable α-amino-3-hydroxy-5-methyl-4-isoxazolepropionic acid (AMPA) receptors, and the increase in Ca^2+^ influx, and metabotropic glutamate receptors. The increase in Ca^2+^ levels leads to the production of toxic substances, such as reactive oxygen and nitrogen species (ROS/RNS), and the activation of Ca^2+^-dependent degradative enzymes [12,13,14,15,16]. In the retina, ischemic conditions that characterize various retinal disorders, including DR, lead to the development of glutamate excitotoxicity and its subsequent deleterious effects on retinal cells [17,18,19]. 

Models of NMDA- or AMPA-mediated excitotoxicity have reported the significant impact of excitotoxicity on different retinal neuronal types, in addition to micro/macroglia activation [20,21,22,23]. Under excitotoxic conditions in the brain, the overactivation of glutamate receptors induces an increase in the activation of NADPH oxidases, including the NOX2 isoform that results in the augmentation of ROS and the development of oxidative stress [24,25,26,27]. In the retina, we showed that AMPA excitotoxicity induced a statistically significant increase in the number of nitrotyrosine-positive (NT^+^) cells (RNS), mainly in the retinal Inner Plexiform Layer (IPL) and Ganglion Cell Layer (GCL) layers [28]. 

Diabetic retinopathy is one of the major complications of diabetes and a neurodegenerative disorder, as it represents the leading cause of preventable blindness in the working-age population and exhibits increasing prevalence globally [29]. In recent years, it has been characterized as a disease of the retinal neurovascular unit, due to diabetes effects on retinal neural and vascular elements [30,31]. A plethora of diverse mechanisms have been associated with the development of DR. Among them, oxidative stress, a result of hyperglycemia-induced dysregulation of different metabolic pathways, is recognized as the intermediate mechanism linking hyperglycemia with the subsequent pathological mechanisms that characterize DR, namely, neuroinflammation, microangiopathy, and neurodegeneration [32]. Mitochondria are considered the initial producers of ROS in the diabetic retina [33], but NADPH oxidases’ activities sustain the oxidative status in the tissue [34].

Glutamate excitotoxicity, as mentioned above, is linked with extensive neuronal loss in DR and neurodegeneration [35]. Elevated levels of glutamate in the retinas of diabetic patients and animal models of DR directly affect retinal neurons [36,37,38]. Oxidative stress is implicated in the increase in glutamate in diabetic retinas [39] due to the dysregulation of glutamate metabolism in retinal macroglia, including both impairments of the glutamate transporter in Müller cells and disrupted enzymatic activity [40,41].

In the present study, we investigated the involvement of NOX2 isoform in pathologies associated with glutamate excitotoxicity and diabetic insults in the retina, by examining the neuroprotective and anti-inflammatory actions of the NOX2-specific inhibitor, GLX7013170, when administered intravitreally (AMPA excitotoxicity model) and topically as eye drops [early-stage diabetic retinopathy (ESDR) paradigm]. GLX7013170 attenuated the AMPA and diabetes-induced insults on neurons and glia. It also reduced the diabetes-induced increase in glutamate levels and partially restored excitatory amino acid transporter (EAAT1) expression. These findings suggest that GLX7013170 administered as eye drops has the pharmacological profile as a potential therapeutic in ESDR, playing a key role in the treatment of the pathologies induced by the triad, oxidative stress, NOX2, and glutamate excitotoxicity.

## 2. Results

### 2.1. Effects of NOX2 Inhibitor in the Experimental In Vivo Model of AMPA Excitotoxicity

#### 2.1.1. NOX2 Inhibition and Retinal Neuroprotection against AMPA Excitotoxicity

Intravitreal administration of AMPA led to a statistically significant reduction in the number of nitric oxide synthetase (NOS)-expressing amacrine cell somata in the Inner Nuclear Layer (INL) and GCL (displaced amacrine cells) and their processes in the IPL. The NOX2 inhibitor, GLX7013170, co-administered with AMPA at a dose of 10^−4^ M (5 μL), partially protected the NOS-positive amacrine cells against AMPA excitotoxicity (Figure 1A,B, *** *p* < 0.001 compared to control, ^##^
*p* < 0.01 compared to AMPA).

#### 2.1.2. NOX2 Inhibition Reduces the AMPA-Induced Activation of Micro/Macroglia

The effect of NOX2 inhibition on retinal glial cells was studied by immunohistochemical studies, using antibodies raised against Ionizing Calcium Binding Adaptor Molecule 1 (Iba-1) and Glial Fibrillary Acidic Protein (GFAP), as specific markers for micro- and macroglia, respectively. 

Microglia: Ιn the AMPA-treated retina, the number of activated Iba-1-positive cells was significantly increased compared to the control tissue. NOX2 inhibition by GLX7013170 (10^−4^ M, 5 μL) reduced the activated Iba-1-positive microglial cells present in the retina (Figure 2A,B, ** *p* < 0.01, *** *p* < 0.001 compared to control, ^##^
*p* < 0.01 compared to AMPA).

Macroglia: AMPA administration caused a statistically significant increase in the expression of GFAP in the retina, an effect that was reversed by co-administration of AMPA with the NOX2 inhibitor, GLX7013170 (10^−4^ M, 5 μL) (Figure 2C,D, * *p* < 0.05 compared to control, ^#^
*p* < 0.05 compared to AMPA).

### 2.2. Effect of NOX2 Inhibitor in the Streptozotocin-Induced Diabetic Retinopathy Model

#### 2.2.1. Expression Profile of NOX2 Isoform mRNA and Effect of NOX2 Inhibition on Oxidative Damage in the Diabetic Retina

Nitrotyrosine, as a result of the nitration of tyrosine residues of proteins due to oxidative stress, is commonly used as a marker for the evaluation of oxidative damage in a tissue. In the two-week model of DR in the present study, diabetes was correlated with a significantly increased number of NT-positive cells present in the retina. Treatment with the NOX2 inhibitor, GLX7013170 [300 × 10^−4^ M (10 mg/mL), 20 μL/eye] reversed the aforementioned effect (Figure 3A,B, *** *p* < 0.001 compared to control, ^###^
*p* < 0.001 compared to diabetic non-treated).

NOX2 isoform is known to be expressed in rat retinas [6]. However, in order to evaluate whether the expression of this specific isoform in the retina is affected in the two-week model of early-stage DR, Real-Time Polymerase Chain Reaction (PCR) analysis was performed. Diabetes was associated with a statistically significant increase in NOX2 mRNA levels in rat retinas, two weeks after the administration of streptozotocin (STZ) (Figure 3C, * *p* < 0.05 compared to control).

#### 2.2.2. Effect of NOX2 Inhibition on the Expression of Bcl-2 Protein in the Diabetic Retina

To evaluate the effect of NOX2 inhibition on apoptotic cell death in the two-week paradigm of DR and the putative anti-apoptotic properties of GLX7013170 (NOX2 inhibitor), we analyzed the expression of the anti-apoptotic protein B-cell lymphoma 2 (Bcl-2) in rat retinas by Western blot. Treatment of the diabetic retina with GLX7013170 led to the reversal of Bcl-2 levels, compared to the diabetic non-treated, suggesting its protective actions to retinal cells against the diabetic insult (Figure 4, * *p* < 0.05 compared to diabetic non-treated retinas).

#### 2.2.3. Role of NOX2 Inhibitor as a Neuroprotectant: NOX2 Inhibition Protects Retinal Neurons

To assess the involvement of NOX2 isoform in the diabetes-induced neurodegenerative changes in the retina and the putative neuroprotective properties of the NOX2 inhibitor, we examined the effect of GLX7013170 on retinal NOS-positive amacrine cells and Retinal Ganglion Cell axons in the ESDR. Retinal Ganglion Cell Axons: The effect of GLX7013170 on the diabetes-induced changes in RGCs was assessed by immunohistochemistry employing Neurofilament (NfL), as a marker for RGC axons. NOX2 blockade did not exhibit any protection against the diabetes insult on RGC axons in the two-week paradigm of DR (Figure 5A,B, * *p* < 0.05, ** *p* < 0.01 compared to control). 

Amacrine cells: Two weeks after the administration of STZ, a significant loss of NOS-positive amacrine cells was observed. Fourteen-day (14 d) treatment with the NOX2 inhibitor, GLX7013170, afforded partial protection of these specific retinal neurons (Figure 5A,C, ** *p* < 0.01 compared to control, ^#^
*p* < 0.05 compared to diabetic non-treated retinas).

#### 2.2.4. NOX2 Inhibition Attenuates the Diabetes-Induced Activation of Micro/Macroglia

Diabetes has been shown to increase the activation of both micro and macroglia. To evaluate the effect of NOX2 blockade, we performed immunohistochemical studies, using Iba-1 and GFAP, as specific markers for microglia and macroglia, respectively.

Microglia: In the two-week paradigm of DR, treatment of the diabetic retina with GLX7013170, reversed the diabetes-induced increase in the number of activated Iba-1-positive cells in the tissue (Figure 6A,B, *** *p* < 0.001 compared to control, ^###^
*p* < 0.001 compared to diabetic non-treated).

Macroglia: Increased GFAP expression was observed in the diabetic non-treated retinas, two weeks after the onset of diabetes. This effect was reduced by the GLX7013170 treatment (Figure 6C,D, * *p* < 0.05, ** *p* < 0.01 compared to control, ^#^
*p* < 0.05 compared to diabetic non-treated).

#### 2.2.5. NOX2 Inhibition Affects Glutamate Metabolism in the Diabetic Retina

Diabetes has been reported to induce the downregulation of several genes related to glutamate metabolism, including glutamate transporters in db/db mice, leading to the accumulation of extracellular glutamate and excitotoxicity [38]. We examined whether diabetes induces changes in glutamate levels and protein levels of glutamate transporter EAAT1 in the two-week model of DR, as well as the effect of NOX2 blockade by GLX7013170. As shown in Figure 7, diabetes led to a statistically significant increase in the levels of glutamate in the diabetic non-treated retinas, an effect that was attenuated by treatment with the NOX2 inhibitor, GLX7013170 (Figure 7A, *** *p* < 0.001 compared to control, ^#^
*p* < 0.05 compared to the diabetic non-treated group).

We further performed Western blot analysis, for the determination of the expression of EAAT1 in the two-week model of DR. Diabetes attenuated EAAT1 levels in the diabetic non-treated retinas. This was partially reversed by treatment with GLX7013170 (Figure 7B, *** *p* < 0.001, **** *p* < 0.0001 compared to control, ^#^
*p* < 0.05 compared to the diabetic non-treated group). 

## 3. Discussion

In the present study, we provide new evidence regarding the involvement of the NOX2 isoform of NADPH oxidases (ΝOΧ1-5) in pathological conditions associated with excitotoxicity and diabetes in rodent retinas. Our findings strongly suggest that the blockade of NOX2, by the selective NOX2 inhibitor, GLX7013170, provides neuroprotection to retinal neurons and glia (micro/macroglia) against AMPA excitotoxicity and diabetic insults. Most importantly, the findings support that NOX2 plays an intermediate role, linking diabetes with the induced alterations in glutamate metabolism that lead to excitotoxicity.

The interplay between glutamate excitotoxicity and oxidative stress has been mainly investigated in studies related to ischemic stroke and brain injury [42]. Under excitotoxic conditions, the increased Ca^2+^ influx induces oxidative stress by the actions of different enzyme systems, including NADPH oxidases, nitric oxide synthase, and xanthine oxidase [43]. In the retina, glutamate excitotoxicity is also correlated with increased ROS production that irreversibly affects the viability and function of retinal neurons and cells [28,44,45].

Following the overactivation of NMDA [24,25] and/or non-NMDA (AMPA, kainic acid) receptors [26,46], different NOX isoforms, including NOX2, have been associated with the overproduction of ROS and subsequent neuronal injury in the brain [24,25,26]. In fact, via the production and extracellular release of superoxide, NOX2-expressing neurons mediate oxidative stress and cell death in neighboring neurons and astrocytes, as a result of NMDA receptor overactivation [47].

### 3.1. NOX2 Inhibition and Neuroprotection against AMPA Excitotoxicity in Retina 

In this study, NOX2 blockade by the selective inhibitor GLX7013170, afforded partial protection to NOS-expressing retinal amacrine cells against excitotoxic insult, caused by the intravitreal AMPA administration. AMPA and other excitatory amino acids (NMDA and kainic acid) are known to severely affect retinal amacrine cells [23,48,49,50]. We have previously shown that inhibition of all NOX isoforms (pan-NOX inhibitor, VAS2870) or specific NOX1 inhibition (ML171) reduced the AMPA-induced loss of NOS-expressing amacrine cells when co-administered with AMPA. In contrast, the NOX4-specific inhibitor (GLX7013114), had no effect [11]. These results suggest that NADPH oxidases are implicated in the AMPA-induced neurodegenerative changes in the retina with NOX isoforms displaying differential roles. 

Glutamate excitotoxicity also affects the levels of micro- and macroglia. In the present study, AMPA administration led to a significant increase in the levels of both micro- and macroglia, as observed by Iba-1 and GFAP immunohistochemical studies, respectively. NOX2 blockade by GLX7013170 attenuated these effects. Similar results have been reported with the inhibition of all NOX isoforms (pan-NOX inhibitor, VAS2870), and the blockade of NOX1 and NOX4 [11], supporting the involvement of NADPH oxidases on the AMPA-induced excitotoxic effect on retinal glia. Our findings are in agreement with Wilkinson-Berka et al. [6] who reported that NOX1, NOX2, and NOX4 isoforms are expressed in retinal micro/macroglia in a model of retinopathy of prematurity. In addition, glutamate excitotoxic insults in brain studies were shown to increase the expression of NOX2 isoform in reactive microglia cells [26]. Thus, excitotoxicity modifies NOX expression on glial cells, affects the activation of micro/macroglia, and regulates the production of extracellular ROS, the expression of pro-inflammatory cytokines, and the subsequent inflammatory response [51]. Micro- and macroglia activation, as a result of the excitotoxic insult, is particularly pertinent, as both glial cell types, once reactive, will induce further damage, consequent neurotoxicity, and retinal cell death [52,53].

### 3.2. Diabetes, Oxidative Stress, Excitotoxicity

Glutamate excitotoxicity is primarily associated with neurodegeneration, but also with neuroinflammation in the CNS (brain and retina). In different ocular disorders, including DR, the function of the glutamatergic system is impaired [54,55], with subsequent effects on retinal cells’ viability and function. Diabetes leads to elevated levels of glutamate in the retina through a series of dysregulations in glutamate metabolism [36,37,38], and changes in the expression of ionotropic glutamate receptors [56], which result in the development of excitotoxicity. These changes have been observed in ESDR [57]. Oxidative stress represents a putative link between diabetes and glutamate metabolism abnormalities in the retina, which are further induced by the excitotoxic insult [37,41,58]. In this circle of events, NADPH oxidases play an important role as a major ROS-producing system. 

### 3.3. Diabetic Retinopathy, Oxidative Stress, and NOX2 Blockade

In the current study, we also examined the expression of NOX2 in ESDR. A significant increase in NOX2 mRNA levels was observed in the retina of diabetic animals, two weeks after STZ administration. The expression of NOX1, NOX2, and NOX4 isoforms of NADPH oxidases has been identified in different types of retinal cells, including macro- and microglia, pericytes, and endothelial cells [6,7], as well as retinal neurons [59]. Our studies are in agreement with other investigations reporting an increased expression of NOX2 in animal models of DR [60,61], as well as an increased expression of other NOX isoforms [9,11,62]. ROS derived from the NOX2 isoform are significantly increased in ESDR, causing mitochondrial damage and further induction of oxidative stress [61]. 

Therefore, increased expression of NOX isoforms, in the retina, will inevitably lead to oxidative damage, which includes an increase in protein nitration. Indeed, in this study, we observed increased nitration of protein tyrosine residues [number of cells positive for the marker nitrotyrosine (NT)] two weeks after the onset of diabetes; an effect that was reversed by treatment with the specific NOX2 inhibitor, GLX7013170. Protein nitration has also been found to be increased in the retinas of diabetic animals in ESDR, such as 15 or 20 days or 4 weeks after the onset of diabetes [11,63,64,65], and in the retinas of human diabetic patients [66]. NT-positive cells were localized in different retinal layers, including the photoreceptor layer, INL and GCL [67]. Based on our results, and the reports of others, we conclude that the NOX2 isoform is implicated in the induction of nitrative stress and tissue damage in the diabetic retina [65,68].

### 3.4. Diabetic Retinopathy, NOX2 Blockade, and Neuroprotection

Diabetes affects retinal neurons leading to cell death [69], with oxidative stress being recognized as one of the primary mechanisms implicated in the neurodegenerative changes that characterize DR [70,71,72]. To examine the effect of NOX2 inhibition in our DR paradigm, we first studied the expression of the anti-apoptotic protein Bcl-2 in the diabetic retina. GLX7013170 reversed the diabetes effect on Bcl-2 levels, suggesting a role for the NOX2 isoform in the regulation of apoptosis in ESDR, in agreement with other studies [61,68,73].

The neuroprotective effect of GLX7013170 on retinal neurons was further evaluated by immunohistochemical studies for NOS-expressing amacrine cells and NfL, a marker for RGC axons. In the ESDR paradigm, diabetes induced a statistically significant reduction in the number of NOS-positive amacrine cells, as well as in NfL immunoreactivity. In line with our data, other studies also observed loss of amacrine cells and NfL immunoreactivity [74,75] and RGCs in the early stages of DR [76,77]. However, NOX2 blockade only partially protected NOS-expressing amacrine cells, while it had no effect on RGC axons. In contrast, previous studies in our laboratory reported that NOX4 inhibition, by the selective inhibitor GLX7013114, in the same two-week DR paradigm, was able to afford full neuroprotection to NOS-positive amacrine cells and to RGC axons [11]. NOX2-derived ROS were also linked to neuronal loss and the induction of neuronal abnormalities in models of ischemia/reperfusion injury [78,79]. Our data suggest that ROS/RNS derived from specific NOX isoforms have differential effects on retinal cell types.

### 3.5. Diabetic Retinopathy, NOX2 Blockade, Anti-Inflammatory Actions

Retinal micro- and macroglia are also affected by diabetes. In agreement with our previous studies, two weeks after the onset of diabetes, a statistically significant increase in the number of activated Iba-1-positive cells and the expression of GFAP protein was observed in the diabetic non-treated retinas [11].

Recruitment and increased activation of microglia in the retina, in ESDR, have also been reported by other researchers [74,80,81]. ROS derived from NOX are shown to induce the activation of microglia and their transition to the pro-inflammatory phenotype [82,83]. The latter leads to the release of pro-inflammatory cytokines, chemokines, and glutamate, as well as ROS, largely derived by NOX isoforms, further affecting the retina [84,85,86].

Increased GFAP expression and the activation of macroglia in the retina have been reported in ESDR animal models [74,81,87,88] and in human diabetic patients with no detectable or non-proliferative DR [89,90]. Macroglia, once activated, produce VEGF, ROS, pro-inflammatory cytokines, and impairments on glutamate metabolism, directly affecting retinal neurons, vascular cells, and the integrity of the blood–retina barrier [31,40,80,84]. In fact, it was shown that activated macroglia, via NOX-induced oxidative damage, affect the survival of retinal neurons [79].

GLX7013170, in our paradigm, significantly attenuated the overactivation of both micro- and macroglia. The involvement of the NOX2 isoform in the mechanisms associated with micro/macroglia activation in the retina has been supported by other researchers, in different models of DR [60,91], and other retinopathies [8]. In light of these data, it appears that NADPH oxidases have a dual role regarding the function of micro/macroglia: NOX-derived ROS from other retinal cells (e.g., neurons, endothelial cells) induce the activation of micro/macroglia, which leads to further production of ROS through NOX-dependent mechanisms, and affect retinal neurons and vascular cells.

The NOX2 isoform has been mainly associated with the vascular impairments that characterize DR [61,91,92,93]. The results of the present study strongly support that in ESDR, NOX2 also plays a key role in neurodegenerative and neuroinflammatory impairments, and that GLX7013170, administered topically, as eye drops, is a potential therapeutic for these impairments.

### 3.6. Diabetic Retinopathy, NOX2 Blockade, and Glutamatergic System

NOX2 was found to be involved in both the effects of AMPA-induced glutamate excitotoxicity and diabetes in the retina. We therefore examined whether this isoform represents a putative link between excitotoxicity and diabetes in our ESDR paradigm. We evaluated the effects of NOX2 blockade on glutamate metabolism and reported a statistically significant increase in glutamate levels in the retinas of diabetic non-treated animals. As mentioned above, an increase in retinal levels of glutamate as a result of diabetes has been identified in many studies, both in diabetic patients and in animal models of DR, as an early diabetic insult [36,37,38,40,87,88,94]. This occurs as a result of various impairments in retinal glutamate metabolism, including alterations in the expression of glutamate transporters. In our model, a significant reduction in the expression of the excitatory amino acid transporter 1 (EAAT1) in the diabetic non-treated retinas was observed. Our data concurred with other studies that also reported similar observations [38,95,96], two weeks after the onset of diabetes [87,88].

EAAT1 represents the predominant glutamate transporter expressed in Müller cells [97], being primarily responsible for the removal of excess glutamate from the extracellular space. Changes in its expression and function are correlated with retinal neuronal loss due to excitotoxicity [98]. In the early stage of diabetes, EAAT1 is found to be vulnerable to oxidative damage, with the impairment of its physiological function [41]. We showed that GLX7013170 partially restores the expression of EAAT1 and, subsequently, reduces glutamate levels in the retinas of diabetic non-treated animals. Thus, oxidative stress, and NOX2 in particular, affects glutamate metabolism in the diabetic retina. Studies in the brain have also identified NOX2 as a mediator for the excessive release of glutamate and the induction of excitotoxicity in pathological conditions such as ischemic stroke [95,99]. The effects of glutamate excitotoxicity are also, to a certain extent, mediated by the induction of oxidative stress. In conclusion, our findings recommend that there is an interplay among the triad, NOX2, oxidative stress, and glutamate excitotoxicity, which promotes neurodegeneration and neuroinflammation in diabetic retinopathy. Selective blockade of the NOX2 isoform protects the retina and suggests GLX7013170, as a neuroprotectant and anti-inflammatory agent, and a potential therapeutic of retinal diseases, including the ESDR.

## 4. Materials and Methods

### 4.1. Animals

Adult Sprague-Dawley rats, both male and female (2–4 months old; 180–300 g) were used in the current study for the induction of two in vivo models, namely, the AMPA-induced excitotoxicity model and the streptozotocin (STZ)-induced diabetic retinopathy model. Animals were maintained under stable housing conditions, at 22 ± 2 °C, on a 12 h light/dark cycle and with free access to food and water. All experiments were performed in accordance with the Statement for the Use of Animals in Ophthalmic and Vision Research and in compliance with Greek national legislation (Animal Act, P.D. 160/91), the EU Directive for animal experiments (2010/63/EU) and the 3Rs principles (i.e., replacement, reduction, refinement). Both animal protocols were approved by the Animal Care Committee assigned by the local Veterinarian Authorities (AMPA excitotoxicity project authorization: 207574, Diabetic Retinopathy project authorization: 207608).

#### 4.1.1. In Vivo Model of AMPA-Induced Excitotoxicity

The in vivo model of AMPA-induced excitotoxicity was utilized as specified by Kiagiadaki and Thermos [23]. Briefly, animals were anesthetized by intraperitoneal administration of a mixture of ketamine (100 mg/kg) and xylazine (10 mg/kg) and placed on a stereotactic device, where intravitreal injections were conducted with the help of a stable flow minipump (Microinjection Pump CMA/100, stable flow rate of 1 μL/min, for 5 min per injection). Each eye received intravitreally a different treatment, which was either PBS (phosphate-buffered saline, 50 mM K2HPO4/ NaH2PO4, 0,9% NaCl, Ph = 7.4), AMPA (Tocris, 8.4 mM, diluted in 50 mM PBS), or AMPA + GLX7013170 (NOX2 inhibitor, 10^−4^ M); thus, there were three distinct experimental groups: control, AMPA, and AMPA treated.

#### 4.1.2. In Vivo Model of STZ-Induced Diabetic Retinopathy

Following an 8–12 h fasting period, a single dose of STZ (70 mg/kg, Sigma-Aldrich, Steinheim, Germany), dissolved in citrate buffer (0.1 M, pH = 4.7) and intraperitoneally administered, was used for the induction of diabetes. Control animals were intraperitoneally injected with 0.1 M citrate buffer. Based on the higher sensitivity to STZ and consequent mortality of male rats [74], the majority of STZ-treated animals were female. All animals with blood glucose levels > 300 mg/dL, 48 **h** after STZ administration, were considered diabetic [74]. On the same day, the diabetic animals were subdivided into two groups, namely, the diabetic non-treated and the diabetic treated groups, and a 14-day treatment with the NOX2 inhibitor, GLX7013170, began (two-week variation paradigm of DR model), and according to the model proposed by Hernandez et al., 2013 [87]. More specifically, the diabetic treated animals received topically, as eye drops (20 μL/eye), GLX7013170 (300 × 10^−4^ M (10 mg/mL), dissolved in DMSO), once daily for 14 days. The diabetic non-treated, as well as the control group, received as eye drops 20 μL of vehicle (DMSO) in each eye, once daily, for the same days of treatment.

### 4.2. Tissue Isolation and Processing

Twenty-four (24) hours after the intravitreal injections in the in vivo model of AMPA excitotoxicity or after the last day (14th) of topical treatment in the in vivo model of DR, animals were euthanized by CO_2_ inhalation and their eyes were enucleated.

#### 4.2.1. Immunohistochemical Studies

Following removal, eyes that would be used in immunohistochemical studies were fixed by immersion in 4% paraformaldehyde (PFA) in 0.1 M phosphate buffer (PB) for 45 min at 4 °C. Subsequently, the posterior part of the eye, including the retina, RPE, choroid and sclera (eyecup) was isolated and repositioned in 4% PFA in 0.1 M PB for 90 min at 4 °C for further fixation. Cryoprotection was achieved by incubation in 30% sucrose buffer in 0.1 M PB overnight at 4 °C. Subsequently, the eyecups were embedded in optimal cutting temperature compound (OCT, VWR Chemicals, Wien, Austria) and rapidly immersed in frozen isopentane for 1 min, at −45 °C and kept at −80 °C till further processing.

Serial, vertical sections of 10 μm thickness were collected consistently from the central retina, using a cryotome (Leica). These were placed consecutively in six, gelatin-covered slides, each one containing eight sections in total. Thus, each slide contained a representative part of the central retina of each retinal sample. Cryostat sections were kept at −20 °C till further use.

#### 4.2.2. Retina Isolation

Immediately after removal, retinal tissue was isolated from the enucleated eyes and kept at −80℃, in order to be further processed in quantitative Real-Time PCR, Western blot analysis, or for the colorimetric quantification of glutamate levels.

### 4.3. Immunohistochemistry

A series of immunohistochemical studies were performed to assess the expression of different markers in the retina in both in vivo models examined. Cryostat sections were washed twice (10 min/wash) with 0.1 M Tris Buffered Saline (TBS, Tris-HCl Ph = 7.4) and incubated for 30 min at room temperature (RT) with blocking buffer, consisting of 3.3% normal goat serum (NGS) in 0.1 M TBS, to block the non-specific binding sites. Following three additional washes with 0.1 M TBS (5 min/wash), sections were incubated overnight at RT with the appropriate primary antibody, diluted in primary antibody buffer containing 0.1 M TBS, 0.5% NGS, and 0.3% Triton X-100. The next day, sections were washed three times with 0.1 M TBS (5 min/wash) and incubated with the proper secondary antibody, diluted in 0.1 M TBS, for 90–120 min, at RT and in dark conditions. After three final 5 min washes with 0.1 M TBS, sections were incubated for 1 min with 4′,6-diamidino-2-phenylindole (DAPI) solution (DAPI diluted 1/2000 0.1 M in TBS), coverslipped with mounting medium (ibidi GmbH, 50001) and kept at 4 °C, in dark conditions till further use. All the antibodies used in the immunohistochemical studies (primary and secondary) are listed in Table 1.

### 4.4. Microscopy and Quantification Studies

Light microscopy images were obtained by the use of a fluorescence microscope Leica DMLB (HCXPL Fluotar, ×20/0.50 or ×40/0.75 lens; Leica Microsystems, Germany), equipped with a Leica DC 300F camera.

The total number of NOS-positive cells (somata) in the inner nuclear layer (INL) and ganglion cell layer (GCL, displaced amacrine cells) was manually counted along the entire length of each retinal section by means of a ×40/0.75 lens. Three sections per slide were counted and the mean was calculated for each slide, representing a different retinal sample.

For the quantification of GFAP and NfL immunoreactivities (IR) the public domain ImageJ (version 1.44) software was employed and the mean gray value [integrated density (fluorescence density)/delineated area] was measured in two pictures taken from the central part of each section, and three different sections were used for each retinal sample (6 images in total/sample). The final measurement for each specific sample was calculated as the mean of the six different values per sample. For GFAP IR, the mean gray value corresponding to the area from GCL to Outer Plexiform Layer (OPL) was measured, while for NfL IR, the measurement included the mean gray value corresponding to the area from GCL to INL.

For the quantification of Iba-1- and NT-positive cells, three different sections were used for each retinal sample and two different pictures were taken from the central part of the retina of each section (3 sections/sample, 2 pictures/section, 6 photos in total/sample). Corresponding DAPI images were taken and the images of Iba-1 or NT IR and DAPI were merged. Subsequently, each merge image was used to manually count the total number of Iba-1- or NT-positive cells, which was then normalized to the total counting area: GCL-INL for Iba-1 IR and GCL-RPE for NT IR. Thus, for each retinal sample there were six different values and mean was calculated in each case.

Crop and rotation of images, as well as adjustments of brightness and contrast were performed using Photoshop CC 2019 (Adobe Systems, San Jose, CA, USA) after the completion of the quantification studies.

### 4.5. Quantitative Real-Time PCR Analysis

Total RNA, isolated from retinal samples based on the TRIzol (Invitrogen, Waltham, MA, USA, catalog 15596026) extraction protocol and the subsequent Turbo DNase (Ambion, Sydney, NSW, Australia, AM2238) treatment, was utilized for cDNA synthesis by the use of the PrimeScript 1st Strand cDNA Synthesis Kit (Takara Bio, Kusatsu, Japan, catalog RR037A), in accordance to the manufacturer’s instructions. KAPA SYBR FAST qPCR Kit Master Mix (2×) (KapaBiosystems, Wilmington, MA, USA, catalog KK4602) and a BIO-RAD CFX Connect Real-Time PCR System were employed for the quantification of the expression levels of NOX2, which were normalized to β-actin, and calculated by the change-in-threshold method; 2–ΔΔCT. The primer sequences for NOX2 and β-actin are in Table 2.

### 4.6. Western Blot Analysis

Retinal samples were homogenized by sonication in lysis buffer containing: 50 mM Tris-HCl pH 7.5, 150 mM NaCl, 1% nonylphenoxypolyethoxylethanol (NP-40), 0.1% deoxycholate (DOC), 0.1 mM phenylmethylsulfonyl fluoride (PMSF), and a protease/phosphatase inhibitor cocktail (Protease: Sigma-Aldrich, S8820/Phosphatase: Merck, 4906845001). Following centrifugation at 10,000× *g* for 20 min, at 4 °C the supernatant was collected and total protein was resuspended in 2X Laemmli sample buffer, consisting of 0.1 M Tris-HCl pH: 7.5, 4% Sodium Dodecyl Sulfate (SDS), 20% glycerol, 0.004% bromophenol blue, and 0.1 M DTT (DiThioThreitol).

Lysates were loaded into a 12,5% acrylamide gel, analyzed by SDS-PAGE and transferred onto nitrocellulose membranes (Macherey-Nagel, Duren, Germany). Following three washes (10 min/wash) with TBS/Tween-20 (TBST), blocking was achieved by incubation in 3% BSA (Sigma-Aldrich, St. Louis, MO, USA) in TBST for 1 h at RT, under agitation. Subsequently, the membranes were incubated overnight with the appropriate primary antibody, diluted in blocking buffer (3% BSA), at 4 °C and under agitation. The next day, membranes were washed again three times (10 min/wash) and incubated with secondary antibodies diluted in blocking buffer, for 1 h at RT and under agitation. The Super Signal West Pico PLUS Chemilluminescent Substrate (Thermo Scientific, Waltham, MA, USA, 34580) was used for protein visualization and ImageJ 1.44 software for the quantification of the optical density of the protein bands. The primary and secondary antibodies used in the Western blot analyses are presented in Table 3.

### 4.7. Glutamate Quantification

Glutamate levels in retinal samples were quantified using a colorimetric Glutamate Assay Kit (abcam, ab83389), following the manufacturer’s instructions. Briefly, retinal tissue was homogenized by sonication in Assay Buffer, incubated for 15 min on ice, and centrifuged at 10,000× *g* for 5 min, at 4 °C. The supernatant was collected and samples, as duplicates, were incubated with the Reaction Mix (containing the Glutamate Enzyme Mix) for 30 min at 37 °C, protected from light, together with freshly prepared glutamate standard samples. The optical density was measured at 450 nm using an ELISA reader (BIO-RAD, model 680), and the concentration of glutamate on each sample was calculated based on the resulting standard curve.

### 4.8. Statistical Analysis

Data were expressed as mean ± SD (standard deviation). GraphPad Prism, version 8.0 (GraphPad Software, San Diego, CA, USA) was used to carry out the statistical analyses, and differences between the experimental groups were assessed either by one-way ANOVA, followed by Newman–Keuls post hoc analysis, or by two-tailed unpaired t-test. The level of statistical significance was set to *p* < 0.05.

## 5. Conclusions

Our findings recommend that there is an interplay among the triad, NOX2, oxidative stress and glutamate excitotoxicity, which promotes neurodegeneration and neuroinflammation in diabetic retinopathy. Selective blockade of the NOX2 isoform protects the retina and suggests that GLX7013170, as a neuroprotectant and anti-inflammatory agent and a potential therapeutic of retinal diseases, including the early stage of diabetic retinopathy.

## Figures and Tables

**Figure 1 pharmaceuticals-17-00393-f001:**
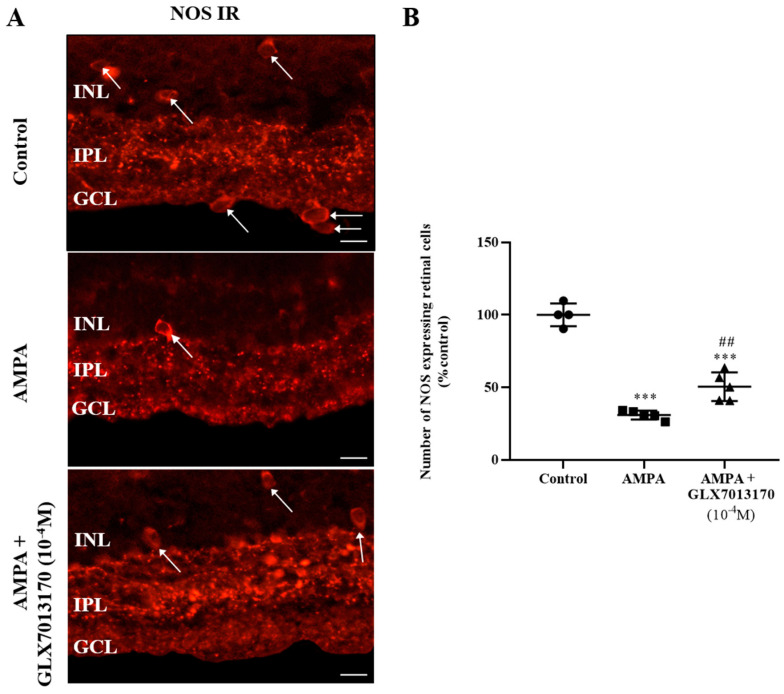
(**A**). Representative images of nitric oxide synthetase (NOS) immunoreactivity (IR) in control, AMPA, and AMPA + NOX2 inhibitor (GLX7013170) rat retinas. Magnification: ×40. Scale bar: 50 μm. Arrows depict NOS-expressing amacrine cells. NOS-positive amacrine cell somata are localized mainly in the Inner Nuclear Layer (INL), processes in the Inner Plexiform Layer (IPL), while displaced amacrine cells are also observed in the Ganglion Cell Layer (GCL). (**B**). Quantification study of NOS IR in control, AMPA, and AMPA + NOX2 inhibitor (10^−4^ M, 5 μL) rat retinas. NOX2 inhibitor, GLX7013170, afforded partial neuroprotection to NOS-positive amacrine cells against AMPA excitotoxicity (*** *p* < 0.001, compared to control, ^##^
*p* < 0.01 compared to AMPA). Data are expressed as Mean ± SD and analyzed by one-way ANOVA, followed by Newman–Keuls post hoc analysis.

**Figure 2 pharmaceuticals-17-00393-f002:**
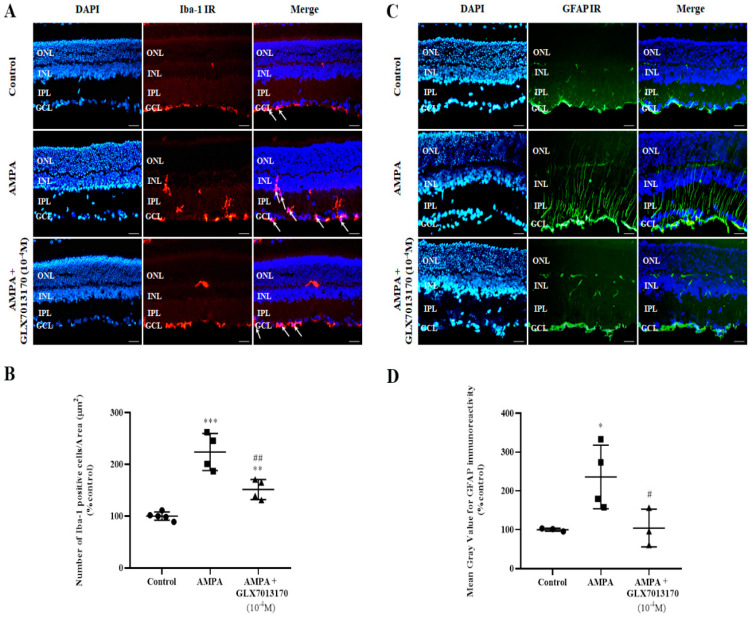
Representative images: (**A**)**.** Ionizing Calcium Binding Adaptor Molecule 1 (Iba-1) IR, (**C**). Glial Fibrillary Acidic Protein (GFAP) IR, respectively, in control, AMPA, and AMPA + NOX2 inhibitor (GLX7013170) rat retinas. Magnification: ×20. Scale bar: 20 μm. Arrows depict Iba-1-positive cells. Corresponding DAPI and Merge images are also presented. Quantification study of Iba-1 IR and GFAP IR, respectively, in the three groups examined. (**B**). Iba-1 IR: The number of Iba-1-positive cells was manually counted from Ganglion Cell Layer (GCL) to Inner Nuclear Layer (INL), followed by normalization to the corresponding area. AMPA administration significantly increased the number of Iba-1-positive cells in the retina, while GLX7013170 limited this effect (** *p* < 0.01, *** *p* < 0.001 compared to control, ^##^ *p* < 0.01 compared to AMPA). (**D**). GFAP IR: Mean gray value of GFAP IR was measured from Ganglion Cell Layer to Outer Nuclear Layer (ONL). AMPA administration led to a statistically significant increase in the expression of GFAP protein in the retina, an effect that was reversed by GLX7013170 (* *p* < 0.05 compared to control, ^#^ *p* < 0.05 compared to AMPA). Data are expressed as Mean ± SD and analyzed by one-way ANOVA, followed by Newman–Keuls post hoc analysis.

**Figure 3 pharmaceuticals-17-00393-f003:**
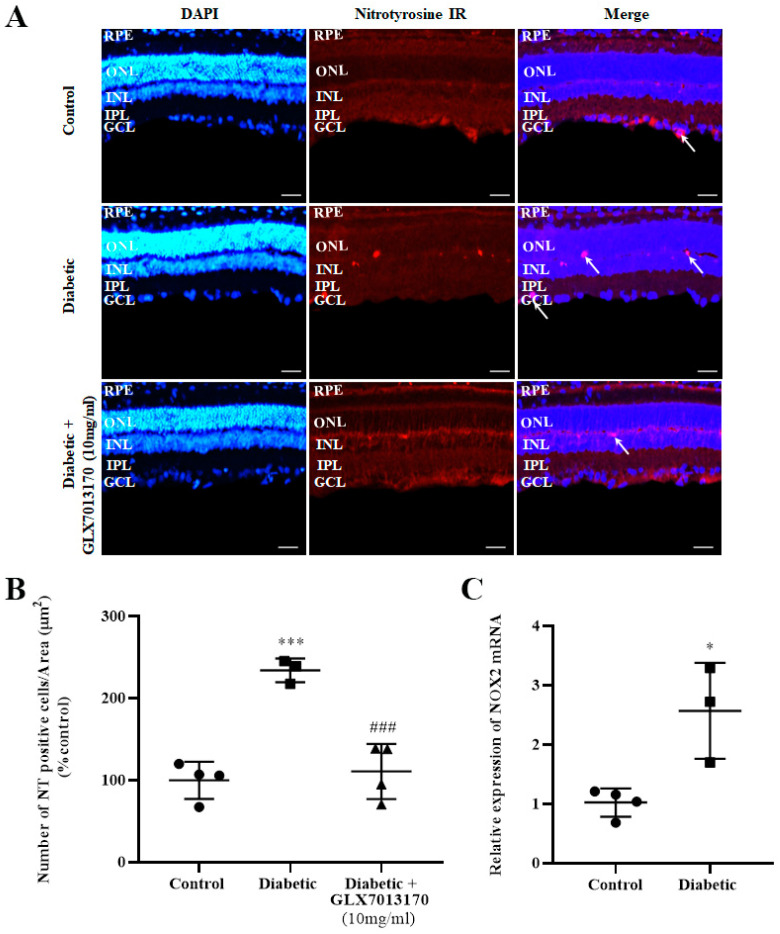
(**A**). Representative images of nitrotyrosine (NT) IR in control, diabetic non-treated, and diabetic treated [GLX7013170, 300 × 10^−4^ M (10 mg/mL), 20 μL/eye] rat retinas, in the 2-week model of DR. Magnification: ×20. Scale bar: 20 mm. Arrows depict NT-positive cells. Corresponding DAPI and Merge images are also presented. (**B**). Quantification study of NT IR in control, diabetic non-treated, and diabetic treated (GLX7013170, 10 mg/mL, 20 μL/eye) retinas. NT-positive cells were manually counted from Ganglion Cell Layer (GCL) to Retinal Pigment Epithelium (RPE) and the number was normalized to the corresponding area. Diabetes significantly increased the number of NT-positive cells in the retina, an effect that was reversed by the NOX2 inhibitor, GLX7013170 (*** *p* < 0.001 compared to control, ^###^
*p* < 0.001 compared to diabetic non-treated). Data are expressed as Mean ± SD and analyzed by one-way ANOVA, followed by Newman–Keuls post hoc analysis. (**C**). Real-Time PCR analysis of the expression of NOX2 isoform in the two-week model of DR. Diabetes caused a statistically significant increase in the mRNA levels of NOX2 isoform in the retina of diabetic non-treated rats, two weeks after the onset of the disease (* *p* < 0.05 compared to control). Data are expressed as Mean ± SD and analyzed by two-tailed unpaired *t*-test.

**Figure 4 pharmaceuticals-17-00393-f004:**
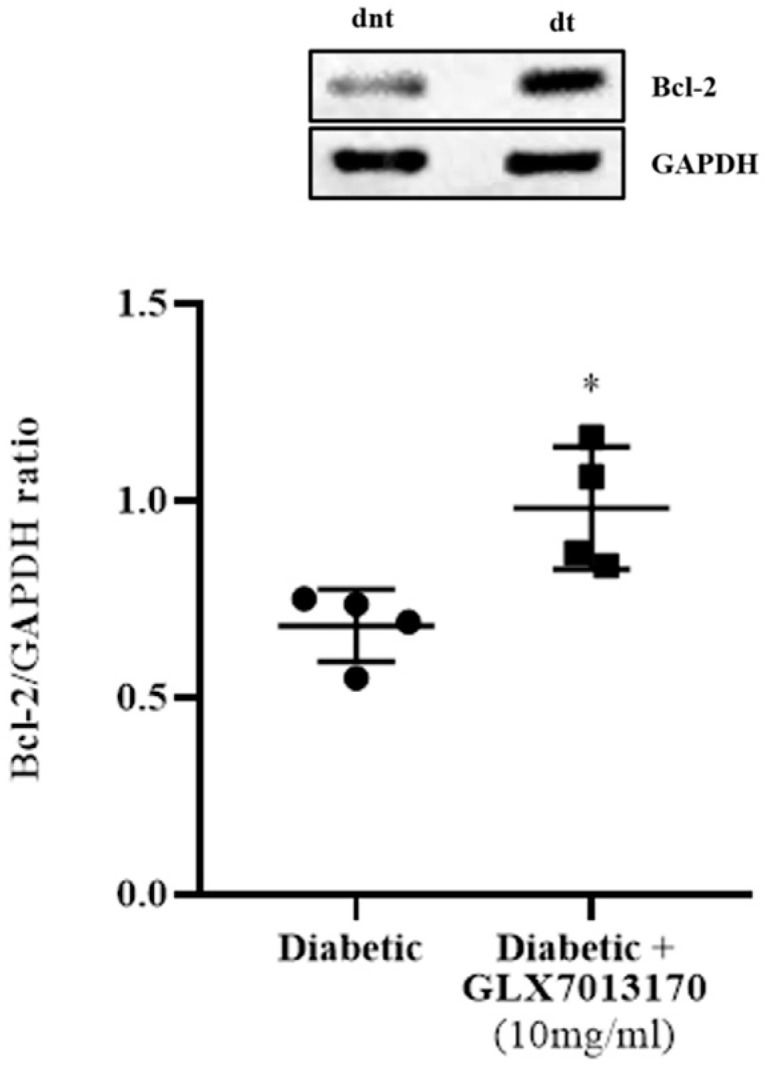
Representative Western blot and quantification study of B-cell lymphoma 2 (Bcl-2) protein expression in control, diabetic non-treated, and diabetic treated [GLX7013170, 300 × 10^−4^ M (10 mg/mL), 20 μL/eye] rat retinas, in the 2-week model of DR. GLX7013170 was associated with the upregulation of Bcl-2 protein in the diabetic retina, two weeks after STZ administration. Data are expressed as Mean ± SD and analyzed by two-tailed unpaired *t*-test. (* *p* < 0.05 compared to diabetic non-treated retinas).

**Figure 5 pharmaceuticals-17-00393-f005:**
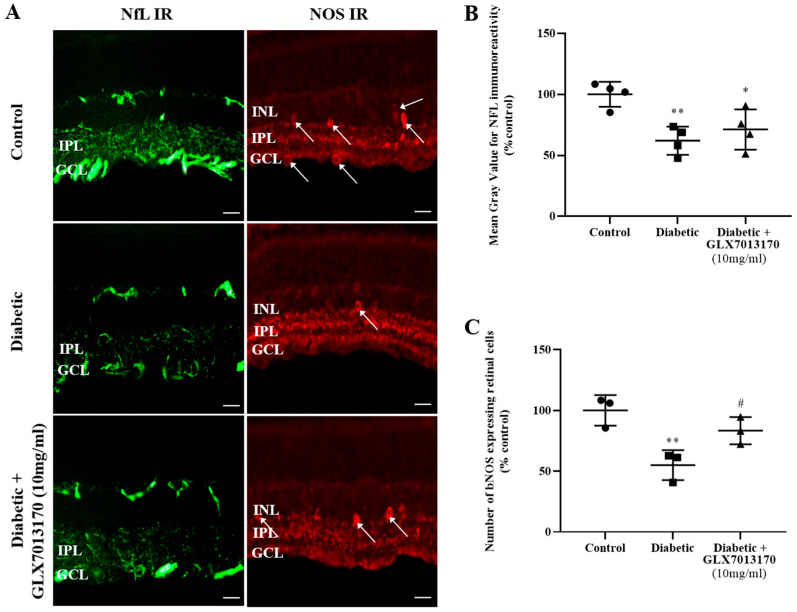
(**A**). Representative images of Neurofilament (NfL) and nitric oxide synthase (NOS) IRs in control, diabetic non-treated, and diabetic treated [GLX7013170, [300 × 10^−4^ M (10 mg/mL), 20 μL/eye] retinas in the two-week DR paradigm. Magnification: ×40. Scale bar: 20 μm. Arrows depict NOS-expressing amacrine cells. (**B**). Quantification study of NfL IR in control, diabetic non-treated, and diabetic treated retinas in the two-week DR paradigm. GLX7013170 had no effect on the diabetes-induced reduction in NfL IR (* *p* < 0.05, ** *p* < 0.01 compared to control). (**C**). Quantification study of bNOS IR in the three groups examined. Diabetes attenuated the number of NOS-positive amacrine cells in a statistically significant manner, an effect that was attenuated by NOX2 inhibitor, GLX7013170 (** *p* < 0.01 compared to control, ^#^
*p* < 0.05 compared to diabetic non-treated). Data are expressed as Mean ± SD and analyzed by one-way ANOVA, followed by Newman–Keuls post hoc analysis.

**Figure 6 pharmaceuticals-17-00393-f006:**
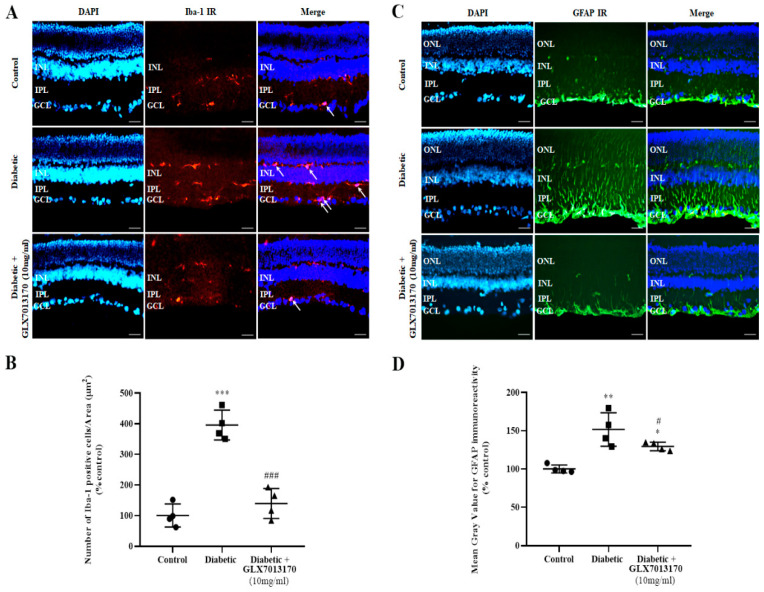
(**A**). Representative images of Iba-1 IR in control, diabetic non-treated, and diabetic treated [GLX7013170, 300 × 10^−4^ M (10 mg/mL), 20 μL/eye] retinas, in the two-week DR paradigm. Magnification: ×20. Scale bar: 20 μm. Arrows depict Iba-1-positive cells. DAPI and Merge are also presented in each case. (**B**). Quantification study of Iba-1 IR in control, diabetic non-treated, and diabetic treated retinas, in the two-week DR paradigm. Iba-1-positive cells were manually counted from Ganglion Cell Layer (GCL) to Inner Nuclear Layer (INL) and the number was normalized to the corresponding area. Diabetes increased the activation of microglia in rat retina, two weeks after streptozotocin administration. GLX7013170 reversed this effect (*** *p* < 0.001 compared to control, ^###^
*p* < 0.001 compared to diabetic non-treated). (**C**). Representative images of GFAP IR in control, diabetic non-treated, and diabetic treated [GLX7013170, 300 × 10^−4^ M (10 mg/mL), 20 μL/eye] retinas, in the two-week DR paradigm. Magnification: x20. Scale bar: 20 μm. DAPI and Merge are also presented in each case. (**D**). Quantification study of GFAP IR in control, diabetic non-treated, and diabetic treated retinas, in the two-week DR paradigm. Mean gray value of GFAP IR was measured from Ganglion Cell Layer (GCL) to Outer Nuclear Layer (ONL). Two weeks after streptozotocin administration, the expression of GFAP in rat retina was significantly increased, while GLX7013170 attenuated this effect (* *p* < 0.05, ** *p* < 0.01 compared to control, ^#^
*p* < 0.05 compared to diabetic non-treated). Data are expressed as Mean ± SD and analyzed by one-way ANOVA, followed by Newman–Keuls post hoc analysis.

**Figure 7 pharmaceuticals-17-00393-f007:**
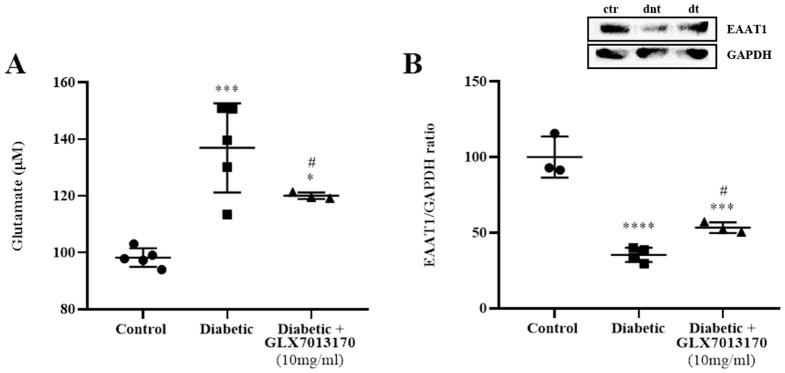
(**A**). Quantification of glutamate levels in control, diabetic non-treated, and diabetic treated [GLX7013170, 300 × 10^−4^ M (10 mg/mL), 20 μL/eye] retinas, in the two-week model of DR. Based on the results of the colorimetric analysis, diabetes significantly increased the levels of glutamate in the retina, two weeks after the administration of streptozotocin. Treatment with the NOX2 inhibitor, GLX7013170, led to the reduction in glutamate levels in the diabetic treated retinas (*** *p* < 0.001 compared to control, * *p* < 0.05 compared to control, ^#^
*p* < 0.05 compared to the diabetic non-treated group). (**B**). Representative Western blot and quantification study of Excitatory Amino Acid Transporter 1 (EAAT1) protein in control, diabetic non-treated, and diabetic treated [GLX7013170, 300 × 10^−4^ M (10 mg/mL), 20 μL/eye) retinas, in the two-week model of DR. The expression of EAAT1 was significantly reduced in the diabetic non-treated retinas, two weeks after the onset of diabetes, an effect that was attenuated by the NOX2 inhibitor, GLX7013170 (*** *p* < 0.001, **** *p* < 0.0001 compared to control, ^#^
*p* < 0.05 compared to the diabetic non-treated group). ctr: control, dnt: diabetic non-treated, dt: diabetic treated. Data are expressed as Mean ± SD and analyzed by one-way ANOVA, followed by Newman–Keuls post hoc analysis.

**Table 1 pharmaceuticals-17-00393-t001:** List of antibodies used in the immunohistochemical studies.

**Primary Antibodies**	**Company/Code/Working Dilution**
Anti-Brain Nitric Oxide Synthetase (bNOS)	Sigma/N7280/1:2000
Anti-Glial Fibrillary Acidic Protein (GFAP)	Sigma/G3893/1:1000
Anti-Ionizing Calcium Binding Adaptor Molecule 1 (Iba-1)	Wako Chemicals/019-19741/1:2500
Neurofilament (NFL)	EMD Millipore/MAB1615/1:500
Nitrotyrosine (NT)	EMD Millipore/06-284/1:1000
**Secondary antibodies**	**Company/Code/Working Dilution**
CF488A goat anti-mouse IgG	Biotium/20010/1:400
CF543 goat anti-rabbit IgG	Biotium/20309/1:1000

**Table 2 pharmaceuticals-17-00393-t002:** Sequences of primers used in Real-Time PCR analysis.

Gene	Forward Primer	Reverse Primer
NOX2	CGCATGCTTTTGAGTGGTTC	AAGTGATTGGCCTGAGATTCATC
β-actin	CTAAGGCCAACCGTGAAAAG	TACATGGCTGGGGTGTTGA

**Table 3 pharmaceuticals-17-00393-t003:** List of antibodies used in Western blot analyses.

Primary Antibodies	Company/Code/Working Dilution
B-cell lymphoma 2 (Bcl-2)	Cell Signaling/2876/1:1000
Anti- EAAT1	Abcam/ab181036/1:1000
GAPDH	Cell Signaling/2118/1:1000
**Secondary antibodies**	**Company/Code/Working Dilution**
HRP-goat anti-mouse IgG	EMD Merck Millipore/AP124P/1:10,000
HRP-goat anti-rabbit IgG	Invitrogen/656120/1:5000

## Data Availability

Data are contained within the article.

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
