# Peer review of "Investigation of the Effects of a Novel NOX2 Inhibitor, GLX7013170, against Glutamate Excitotoxicity and Diabetes Insults in the Retina"

_pharmaceuticals, 2024, doi:10.3390/ph17030393_

Round 1

Reviewer 1 Report

Comments and Suggestions for Authors

The study investigating the effects of the NOX2 inhibitor GLX7013170 on glutamate excitotoxicity and diabetic retinopathy is interesting, but several concerns need to be addressed before publication.

1. GLX7013170 dosage: Please provide details on how the GLX7013170 doses were chosen for this study. Justify the selection based on previous literature or preliminary experiments.

2. Diabetic Retinopathy models:

  • Phenotype characterization: Please describe the established phenotypes of your AMAP and streptozotocin-induced diabetic retinopathy models in detail. Include quantitative data on neuron number changes using established markers like NeuN staining. Additionally, strengthen your glial activation data by providing images with higher magnification or using more specific markers.
  • DR Phenotype in AMPA Model: The limited NT positive cells in Figure 2 raise concerns about the robustness of the AMPA-induced DR phenotype.

3. Neuron numbers: Quantify neuron number changes in non-treated and GLX-treated retinas in the AMPA model using NeuN staining or similar methods.

4. Anti-apoptotic data: While Bcl-2 data is helpful, consider including TUNEL staining to provide a more comprehensive picture of apoptosis. Regarding the Western blot original image of Figure 4, ensure all lanes are clearly labeled, including controls. Explain what each lane represents and provide a complete blot image showing the control band in figure 4.

5. NOS staining: The apparent discrepancy between the visual increase in NOS staining and the statistically significant decrease requires clarification. Consider re-evaluating the staining or providing a detailed explanation for the observed discrepancy. Additionally, improve the image quality by increasing magnification.

6. Western blots: Enhance the image quality of the EAAT1 and GFAP Western blots in Figure 7 for better analysis. The original image of EAAT1 Western blot quality is not good.

Comments on the Quality of English Language

Additional recommendations:

  • Proofread the manuscript for any grammatical or typographical errors.

Reviewer 2 Report

Comments and Suggestions for Authors

In this manuscript, Stavroula Dionysopoulou et al. conducted a systematic study on a novel NOX2 inhibitor, GLX7013170. Their findings revealed that this compound can effectively protect the retina from oxidative stress, NOX, glutamate excitotoxicity, etc. The manuscript contains a wealth of experimental results, demonstrating a comprehensive and well-thought-out experimental design. However, the manuscript needs to be revised carefully due to numerous errors and typos.

1. Abbreviations should be expanded to their full form the first time they appear in the manuscript. For example, the “RGCs” in line 55 should be corrected to “retinal ganglion cells (RGCs)”. A similar approach should be applied to abbreviations such as NMPA (line 67), AMPA (line 68), IPL (line 80), etc. Alternatively, a table of abbreviations can be introduced at the beginning of the manuscript to avoid confusion.

2. The manuscript exhibits inconsistency in metric units. The dosing unit of GLX7013170 in the 2.1.1 section is 10e-4M, whereas in subsequent sections, it switches to 10mg/mL. It is highly recommended to harmonize the metric unit so that readers can easily compare the dosages. Moreover, it is crucial to present the complete dosage rather than just the concentration. For example, it is recommended to express the GLX7013170 dose as "GLX7013170, 10mg/mL, 20uL/eye" instead of "GLX7013170, 10mg/mL" as observed multiple times in the manuscript. This enhancement provides a clearer and more informative representation of the dosage.

3. A careful check for typos is essential. Examples include missing spaces, such as in "EAAT1expression" (line 106), the loss of superscripts and subscripts in lines 483 to 485, and correcting the dosage in line 499 to "20uL/eye" instead of "20mL/eye." These errors diminish the scientific clarity of the manuscript.

Round 2

Reviewer 1 Report

Comments and Suggestions for Authors

The authors answered my questions. I have no more. Thanks.